# Unveiling Angiotensin II and Losartan-Induced Gene Regulatory Networks Using Human Urine-Derived Podocytes

**DOI:** 10.3390/ijms241310551

**Published:** 2023-06-23

**Authors:** Chantelle Thimm, Lars Erichsen, Wasco Wruck, James Adjaye

**Affiliations:** 1Institute for Stem Cell Research and Regenerative Medicine, Medical Faculty, Heinrich-Heine University Düsseldorf, 40225 Düsseldorf, Germany; chantelle.thimm@uni-duesseldorf.de (C.T.); lars.erichsen@uni-duesseldorf.de (L.E.); wasco.wruck@med.uni-duesseldorf.de (W.W.); 2EGA Institute for Women’s Health, Zayed Centre for Research into Rare Diseases in Children (ZCR), University College London (UCL), 20 Guilford Street, London WC1N 1DZ, UK

**Keywords:** urine, podocytes, disease modeling, renin–angiotensin–aldosterone system, kidney, losartan, hypertension

## Abstract

Podocytes are highly specialized cells that play a pivotal role in the blood filtration process in the glomeruli of the kidney, and their dysfunction leads to renal diseases. For this reason, the study and application of this cell type is of great importance in the field of regenerative medicine. Hypertension is mainly regulated by the renin–angiotensin–aldosterone system (RAAS), with its main mediator being angiotensin II (ANG II). Elevated ANG II levels lead to a pro-fibrotic, inflammatory, and hypertrophic milieu that induces apoptosis in podocytes. The activation of RAAS is critical for the pathogenesis of podocyte injury; as such, to prevent podocyte damage, patients with hypertension are administered drugs that modulate RAAS signaling. A prime example is the orally active, non-peptide, selective angiotensin-II-type I receptor (AGTR1) blocker losartan. Here, we demonstrate that SIX2-positive urine-derived renal progenitor cells (UdRPCs) and their immortalized counterpart (UM51-hTERT) can be directly differentiated into mature podocytes. These podocytes show activation of RAAS after stimulation with ANG II, resulting in ANG II-dependent upregulation of the expression of the angiotensin-II-type I receptor, AGTR1, and the downregulated expression of the angiotensin-II-type II receptor 2 (AGTR2). The stimulation of podocytes with losartan counteracts ANG II-dependent changes, resulting in a dependent favoring of the specific receptor from AGTR1 to AGTR2. Transcriptome analysis revealed 94 losartan-induced genes associated with diverse biological processes and pathways such as vascular smooth muscle contraction, the oxytocin signaling pathway, renin secretion, and ECM-receptor interaction. Co-stimulation with losartan and ANG II induced the exclusive expression of 106 genes associated with DNA methylation or demethylation, cell differentiation, the developmental process, response to muscle stretch, and calcium ion transmembrane transport. These findings highlight the usefulness of UdRPC-derived podocytes in studying the RAAS pathway and nephrotoxicity in various kidney diseases.

## 1. Introduction

Kidneys perform numerous essential tasks in the human body; these are filtration of the urine to dispose of urinary components such as urea, uric acid, and creatine into the urine [1]. They regulate the homeostasis of electrolytes such as sodium, potassium, calcium, magnesium, and phosphate. Additionally, kidneys adjust the acid–base balance and control blood pressure by regulating the synthesis of calcitriol, erythropoietin, and renin [1].

Podocytes are highly differentiated and specialized cells located on the outside of the glomerular capillaries that project into the urinary cavity of Bowman’s capsule. They have a large cell body from which numerous foot processes taper around the glomerular capillaries. Adjacent podocytes thus form lamellipodia-like foot processes, between which an interdigitating cell–cell contact is formed to ensure the size selectivity of glomerular filtration via the slit membrane. Numerous podocyte-specific genes have been identified whose products play important roles in maintaining the filtration barrier. There is a highly conserved complex consisting of Podocin (*NPHS2*), Nephrin (*NPHS1*), KIRREL1 (*NEPH1*), and CD2-associated protein (*CD2AP*) through which podocytes are connected to cell–cell contacts and control podocyte function [2,3,4]. The morphology of podocytes, which form the filtration slits with their primary and secondary foot processes, can only be maintained by a mature internal support network.

The renin–angiotensin–aldosterone system (RAAS) is a cascade that contributes to various systematic and localized processes. Among the systematic processes is the regulation of blood pressure. The octapeptide angiotensin II (ANG II) is the biological effector of the renin–angiotensin system and regulates blood pressure and extracellular volume. Angiotensin II acts on the two subtypes of angiotensin II receptors AT1 (AGTR1) and AT2 (AGTR2). Based on this, AGTRs were shown to be expressed in various segments of the nephron in the distal tubule, collecting duct, and renal vasculature [5,6,7,8,9,10]. The binding of ANG II to the AGTR1 activates four classical signaling cascades, including phospholipase A2, phospholipase C, phospholipase D, and L-type calcium channels, and normally blocks the adenylate cycle [11,12]. Stimulation of these signaling pathways via renal AGTR1 initiates vasoconstriction, sodium re-absorption, protein synthesis, and cellular growth [13,14]. AGTR2 receptors are comparable to AGTR1 receptors in their structure, tissue expression, signaling pathways, and functions. The activation of AGTR2 mediates K+ channel activity via the G protein G1 and activates protein tyrosine phosphatase (PTP), which leads to a reduction in MAPK- activity or ERK1 [15,16]. It should be noted that in the human kidney, the expression of AGTR1 is 8–10-fold higher at the mRNA level compared to AGTR2. Moreover, the activation of AGTR2 leads to increased bradykinin production, which induces vasodilation via the NO/cGMP pathway [13].

Persistent activation of the RAAS is critical for the pathogenesis of podocyte injury and causes proteinuria. ANG II stimulation triggered the downregulation of *ACTN4* and decreased expression of focal adhesion in podocytes [17]. In summary, with sustained RAAS activation, the phenotype induced by ANG II shifts from dynamic stability to adaptive migration, which can eventually lead to podocyte depletion with high actin cytoskeletal turnover and consequent podocyte depletion and focal segmental glomerulosclerosis [18,19]. The beneficial effects of RAAS modulation on proteinuria reduction, whether occurring via ACE inhibitors (e.g., Ramipril, Benazapril, and Captopril) or AGTR1 antagonists (e.g., losartan, Candesartan, and Irbesartan), are confirmed by clinical evidence [20,21]. ACE inhibitors and AGTR1 antagonists are these days indispensable in the management of hypertension and blood pressure. In the 1990s, the development of orally active, non-peptide, selective AGTR1 blockers began with the synthesis of losartan [22]. In contrast to the application of ACE inhibitors, AGTR1 blockers do not prevent the formation of ANG II, but rather the binding of the peptide to AGTR1. With respect to AGTR1 binding inhibition, the increase in ANG I leads to an increase in ANG II levels, which can bind freely to AGTR2 or other receptor subtypes [14,23].

We previously reported human urine as a non-invasive source of renal stem cells with regenerative potential. These urine-derived renal progenitor cells (UdRPCs) express, among others, the renal stem cell markers sine oculis homeobox homolog 2 (SIX2), Cbp/P300-interacting transactivator with Glu/Asp rich carboxy-terminal domain 1 (CITED1), Wilms tumor 1 (WT1), CD133, CD24, and CD106 [24]. Recently, we established a protocol that leads to the differentiation of human UdRPCs into mature podocytes with typical cellular processes. We reported the full characterization of the generated podocytes at the transcriptome, secretome, and cellular levels [19]. Furthermore, we showed that ANG II-stimulation of the podocytes activated the renin–angiotensin aldosterone system, resulting in a downregulated expression of NPHS1 [19]. This subsequently leads to cytoskeletal changes and increased AGTR1 expression and downregulation of AGTR2 expression.

In the current study, we demonstrate that losartan counteracts ANG II-dependent molecular changes by elevating the expression of podocyte-specific markers such as NPHS1. Additionally, cells treated simultaneously with ANG II and losartan undergo receptor-usage switching from AGTR1 to AGTR2. We carried out a transcriptome analysis of human urine-derived podocytes stimulated with 100 μM ANG II with and without 1 μM losartan. To date, transcriptome data derived from podocytes treated with losartan are restricted to rodents [25,26,27]. Our current data re-affirms that human urine-derived podocytes are a valuable in vitro model for dissecting RAAS, studying podocyte-associated diseases, and drug screening, thus complementing the use of pluripotent stem cells. 

## 2. Results

### 2.1. Losartan Rescues ANG II Induced Down-Regulation of NPHS1 Expression

In a previous manuscript, we reported the existence of urine-derived renal progenitor cells (UdRPCs) inherently capable of differentiating into podocytes [19,24,28]. UdRPCs adopt the typical “fried egg” morphology typical of podocytes after seven days of continuous culturing in adv. RPMI medium supplemented with 30 μM retinoic acid (RA). We previously showed the expression of foot processes and slit-diaphragm-associated proteins- NPHS1 [19]. Furthermore, because these cells have been shown to be responsive to angiotensin II (ANG II) [20], we applied the angiotensin II receptor type I blocker losartan to investigate its protective effect on UdRPC-derived podocytes treated with ANG II.

For the investigation, the optimal concentration of losartan was tested with and without ANG II. Based on these findings, a losartan concentration of 1 μM and a cell density of 60–70% were used in all further experiments. Therefore, the urine-derived podocytes were pre-treated with 1 μM losartan for 24 h, thereafter 100 μM ANG II was added for another 24 h. 

We differentiated primary SIX2-positive UdRPC UM51 and the transformed counterpart UM51-hTERT into podocytes [28]. The expression of NPHS1 was analyzed to confirm the differentiation of both cell types into mature podocytes, and phalloidin staining was employed to reveal cytoskeletal morphologies (Figure 1A,B). The addition of ANG II for 24 h induced rounded-up morphological changes in some cells, but all cells expressed NPHS1 (Figure 1A,B). 

### 2.2. The Effect of Angiotensin II and Losartan on the Expression of Angiotensin Receptor Type 1 and 2

High ANG II concentrations impart a strong vasoconstriction effect that is associated with an increase in blood pressure. To assess the effect of ANG II and the ability of the generated podocytes to model RAAS with the involvement of AGTR1 and AGTR2, a final concentration of 100 μM ANG II in adv. RPMI supplemented with 30 μM RA for 24 h was used to treat the cells. The gene-specific mRNA expression of AGTR1, AGTR2 receptors, and NPHS1 were analyzed via qRT-PCR (Figure 2A–C). The expression of AGTR1 tended to be upregulated by 24 h of ANG II treatment in the UM51 podocytes and UM51 hTERT podocytes. In the UM51 podocytes, we observed a significant increase in the AGTR1 expression after ANG II treatment (*p* = 0.002). In contrast, the addition of 1 μM losartan induced a slight significant downregulation of AGTR1 (*p* < 0.05) compared to ANG II treatment (Figure 2A). The expression of AGTR2 was significantly downregulated after 24 h of ANG II treatment in UM51 (*p* = 0.007) while in UM51, hTERT podocyte mRNA expression was slightly increased (Figure 2B).

Furthermore, the expression of the podocyte structural protein NPHS1 was analyzed using RT-PCR and Western blotting. We show that NPHS1 expression is significantly downregulated in the UM51 hTERT podocytes by ANG II treatment for 24 h (*p* = 0.02) (Figure 2C). As hypothesized, this transcriptional downregulation was counteracted by losartan, leading to a significant increase in NPHS1 (*p* < 0.05) expression in both cell lines (Figure 2C). To further confirm the effect of losartan, the protein expression of AGTR1, AGTR2, and NPHS1 were determined using Western blot analysis. ANG II treatment increased AGTR1 expression and induced a concomitant downregulation of AGTR2 (Figure 2D). This effect is counteracted by the addition of losartan, resulting in a significant decrease in AGTR1 (*p* = 0.004 and *p* = 0.01) expression compared with ANG II treatment (Figure 2D). In parallel, AGTR2 expression significantly increased compared to ANG II treatment (*p* = 0.003 and *p* = 0.004) (Figure 2D). We propose that losartan modulates the expression of both receptors, with a bias, however, towards the AGTR2 downstream signaling pathway.

### 2.3. Comparative Transcriptome Analysis of Urine-Derived Podocytes Treated with and without Losartan

The podocytes were treated with 100 μM ANG II or 1 μM losartan or the combination of 100 μM ANG II and 1μM losartan, and we performed a comparative transcriptome analysis. Hierarchical clustering analysis comparing the transcriptomes revealed a distinct expression pattern for all treatments compared to the untreated control podocytes (Appendix A). A total of 13,413 genes were detected as expressed in both the untreated and 1 μM losartan treated podocytes. Comparing the expressed genes (det-*p* < 0.05), 1219 are exclusively expressed in the untreated control podocytes and 94 in the podocytes treated with losartan alone (Figure 3A). The most over-represented GO BP-terms exclusive to untreated podocytes are associated with the regulation of system processes, epithelial cell differentiation, negative regulation of cell population proliferation, and cytokine–cytokine receptor interaction (Appendix A, sheet 1). In comparison, podocytes treated with 1 μM losartan showed the most over-represented GO BP-terms associated with regulation of monoatomic ion transmembrane transporter activity, cell morphogenesis involved in neuron differentiation, and cell–cell communication (Figure 3B,D). The significant genes and associated KEGG pathways revealed vascular smooth muscle contraction, the oxytocin signaling pathway, renin secretion, and ECM-receptor interaction (Figure 3C). The entire KEGG pathway is presented in Appendix A and Appendix A. Figure 3D represents an overview of the exclusive GO BP-terms for losartan alone used in this study. The full gene lists can be found in Appendix A.

### 2.4. Comparative Transcriptome Analysis of Urine-Derived Podocytes Treated with Angiotensin II and the Combination of Angiotensin II and Losartan

We detected 13,131 genes as co-expressed between podocytes treated with 100 μM ANG II and the combination of 100 μM ANG II and 1 μM losartan (Figure 4A). In total, 1243 genes were exclusively expressed in podocytes treated with ANG II alone. In the combination of ANG II and losartan, 106 unique genes were expressed (Figure 4A). The most over-represented GO BP-terms exclusive to podocytes treated with 100 μM ANG II are associated with cell division, regulation of cell cycle process, renal system development, metal ion transport, and regulation of hormone levels (Appendix A, sheet 4). In comparison, podocytes treated with the combination of 100 μM ANG II and 1 μM losartan showed the most over-represented GO BP-terms associated with extracellular matrix organization, metal ion transport, DNA methylation or demethylation, cell differentiation, developmental process, response to muscle stretch, calcium ion transmembrane transport, and regulation of cytokine production involved in inflammatory response (Figure 4B,D). The significant genes and associated KEGG pathways revealed including protein digestion and absorption, pancreatic secretion, and primary immunodeficiency, amongst others. (Figure 4C). The entire KEGG pathway is presented in Appendix A.

To summarize, our results provide detailed information on ANG II and losartan-induced gene expression and associated signaling pathways. The full gene lists can be found in Appendix A.

## 3. Discussion

The epithelial cells of the glomerulus called podocytes are highly differentiated and specialized cells that are located on the outside of the glomerular capillaries and project into the urinary cavity of Bowman’s capsule. They have a large cell body from which numerous foot processes taper around the glomerular capillaries. Worldwide, 1.28 billion adults aged 30–79 years are afflicted with elevated blood pressure [29]. Uncontrolled high blood pressure causes serious damage to the heart and kidneys that eventually leads to death. RAAS represents a key mechanism in the regulation of blood pressure and the electrolyte and fluid balance of the organism. Activation of the RAAS is mediated by two receptors, angiotensin II receptor type I and II. Studies suggest that AGTR1 in the kidney is responsible for hypertension in mammals [30]. Based on this, AGTRs were shown to be expressed in various segments of the nephron in the distal tubule, collecting duct, and renal vasculature [5,6,7,8,9,10]. The stimulation of AGTR1, which is expressed 8–10 fold higher at the mRNA level than AGTR2 in the human kidney, is accompanied by renal vasoconstriction, which consequently impairs the capacity of the kidney to process sodium by decreasing medullary blood flow [31]. Additionally, other groups have reported that ANG II can induce damage to and the subsequent loss of podocytes [32]. AGTR1 can be activated by the main mediator of the RAAS, namely ANG II. An increased amount of circulating and intracellular ANG II and aldosterone causes a pro-fibrotic, -inflammatory, and -hypertrophic milieu that triggers the remodeling and dysfunction of the podocyte cytoskeleton.

To date, human induced pluripotent stem cells (iPSCs) derived from somatic cells are available and serve to study nephrogenesis and various kidney associated diseases such as hypertension [33,34,35,36,37]. In the present study, we investigated gene-regulatory networks induced by ANGII and losartan employing our previously established podocyte differentiation protocol based on urine-derived nephron progenitor cells (UdRPCs) [19,28]. We specifically modeled the RAAS pathway using our established primary SIX2-positive UM51 and the transformed counterpart UM51-hTERT cell line. In our previous work, immunofluorescence-based protein detection, Western blotting, and RT-PCR were used to observe a change in the podocyte cytoskeleton after 24 h of ANG II treatment [19]. ANG II treatment led to the alteration of the podocyte-specific fried-egg morphology to a rounded-cell morphology. It is known that downregulation in the expression of podocyte-specific genes, such as NPHS1, is associated with podocyte injury [38]. Consequently, there is a disturbance of the foot processes and the slit diaphragm [39,40]. An upregulation in angiotensin II receptor type I expression was shown after 24 h treatment with ANG II. The biological function of this receptor is associated with the mediation of vasoconstriction, cell proliferation, nephrosclerosis, hypertrophy of vascular media, endothelial dysfunction, inflammation and immune responses, and promotion of aging [11,41]. It is questionable whether podocytes can adapt their phenotype and whether the observed changes even remain chronic when stimulated with ANG II over a longer period. Currently, numerous angiotensin receptor blockers (ARB) and angiotensin converting enzyme inhibitors (ACEI) are used in the clinic to manage a variety of renal disorders; one of these ARBs is losartan. It is an anti-hypertensive agent from the group of sartans used to treat hypertension, heart failure, and diabetic nephropathy. Its action is based on reversing the effects of ANG II at AGTR1 by blocking AGTR1 [42]. ANG II signaling through AGTR2 has been associated with vasodilatation, development, cell differentiation, tissue repair, and apoptosis [11].

The use of losartan allowed us to determine whether the cytoskeletal change was mediated by ANGII induced signaling through AGTR1. The present data indicate that the addition of losartan is sufficient to counteract the cytoskeletal restructuring induced by ANG II by rescuing NPHS1 expression. It is interesting to note that losartan causes a redistribution of the expression of AGTR1 and AGTR2. From this, it can be concluded that losartan can efficiently block binding to AGTR1, causing ANG II to bind with a higher preference, presumably to AGTR2. Furthermore, this indicates that activated angiotensin receptors trigger a signaling cascade into the nucleus to augment their own transcriptional and translational expression. AGTR2 appears to reintroduce cells into differentiation, again upregulating the expression of podocyte-specific genes. 

The 1243 genes that were upregulated in the podocytes treated with ANG II are annotated with GO-BP terms such as renal system development, cell division, metal ion transport, calcium signaling pathway, and regulation of the cell cycle process, suggesting that these cells might dedifferentiate and lose their cell type-specific properties by signaling through AGTR1. Podocytes are terminally differentiated cells that do not undergo cell division in vivo. Therefore, a process of dedifferentiation is needed when kidney damage occurs. This process has been described for many renal pathologies, such as diabetes and lupus nephritis, and is even discussed as a potential biomarker [43]. In our model, we hypothesize that this dedifferentiation might reflect the need for highly differentiated podocytes to re-enter the cell cycle to regenerate tissue because of ANGII-induced damage. 

In comparison, GO-BP terms for losartan treatment included cell–cell communication, vascular smooth muscle contraction, oxytocin signaling pathways, and renin secretion. These include the calmodulin-like 4 (CALML4), natriuretic peptide A (NPPA), and protein phosphatase 1 regulatory subunit 12B (PPP1R12B) genes. The blockade of AGTR1 and the resulting signaling through AGTR2 seem to lead to vasodilation and reduction in vascular tone [44,45,46]. In addition, metal ion transport seems to play an essential role in the distinct signaling cascades of the receptors and the conformation of ANGII itself [10,47]. More precisely, the interaction of metal ions with the imidazole moiety of His-6 of ANGII has been shown to influence the conformation of human ANG II [47]. ANG II-stimulated podocytes show upregulated expression of genes including adrenoceptor alpha 1A (*ADRA1A*) and shroom family member 2 (*SHROOM2*). Like AGTR1, the alpha-adrenergic receptor *ADRA1A* mediates its action through association with G-proteins that activate a phosphatidylinositol calcium second messenger system. These in turn activate mitogenic responses and regulate the growth and proliferation mediated by Klf15 in the kidney [48]. Additionally, the loss of foot processes may be explained by *SHROOM2*, which is involved in endothelial cell morphology changes during cell spreading [49]. 

Some of the losartan-induced 94 genes are associated with the GO-BP term metal ion transport, which includes *LCK*, *NPPA*, *CCL3*, *SLC10A1*, *SLC12A3*, *CACNA1H*, *CATSPER1*, *LRRC38*, *HEPHL1*, *CFTR*, *TMC8*, *CSRP3*, *DMD,* and *MMP9.* These genes have been reported to contribute to reduction in inflammation and oxidative stress [50]. ARBs prevent abnormal iron deposition in the interstitium, correct chronic hypoxia, reduce the expression of heme oxygenase and p47phox, and inhibit pentosidine formation [51]. *NPPA*, for example, is responsible for extracellular fluid volume and electrolyte homeostasis. This signaling cascade can apparently be activated by the addition of losartan and seems to play a major role in hypertension [45,52,53]. The inhibition of tyrosine kinase activity by *LCK* results in a reduction in oxidative stress as described by Alqarni [54]. Interestingly, the inhibition of metal ions catalyzes oxidation of ascorbic acid, and abnormal depositing of metal ions in the interstitium is unique to ARBs, but not to calcium channel blockers (CCBs) and ß blockers (BBs). This might imply superior renal protective effects of ARBS related to the prevention of oxidative stress.

The 106 genes associated with the co-stimulation of losartan and ANG II are annotated with GO-BP terms such as extracellular matrix organization, metal ion transport, DNA methylation or demethylation, cell differentiation, developmental process, response to muscle stretch, regulation of cytokine production involved in inflammatory response, and calcium ion transmembrane transport. It has already been reported that podocytes expresse the transient receptor proteins TRPC3, TRPC4, TRPC5, and TRPC6 [55,56]. Under normal conditions, these receptors mediate calcium influx into podocytes, by which they can adapt their cytoskeleton to environmental cues. This actin remodeling is Ca^2+^ dependent. As early as 1978, Kerjaschik was the first to propose that the Ca^2+^ increase in podocytes is one of the first events leading to podocyte injury [57]. AGTR1 serves as an upstream receptor for both TRPC5 and TRPC6 in podocytes [58]. It was previously suggested that the activation of Calcineurin appears to be downstream of the AGTR1 signaling pathway via ANG II [59,60]. Upstream of AGTR1, G protein-coupled receptors activate phospholipase C, which grants Ca^2+^ influx via TRPC5/TRPC6. This cascade enables the downstream transcription of Calcineurin. AGTR1, TRPC, Calcineurin, and the actin cytoskeleton form a linear signaling cascade in which ANG II activates Calcineurin in a TRPC5-dependent manner [58]. Changed calcium influx leads to the loss of podocyte foot processes and the downregulation of SYNPO as well as TRPC6 [61]. This downregulation of the podocyte-specific marker SYNPO was already demonstrated in one of our previous manuscripts [28]. The effect of ANG II on human urine-derived podocytes is also reflected in our data, in which calcium signaling is associated with ANG II treatment. ANGII induced activation of AGTR1 signaling is reflected by the biological effects of AGTR1, which initiates proliferation as well as calcium-dependent changes in cellular morphology. Another possible mode of action was revealed by a study carried out by Schenk et al. [62]. Here, the authors showed the protein expression and phosphorylation of LCP1 in ANGII treated podocytes through AGTR1 activation [62]. It is known that LCP1 is involved in actin-bundling and that this function is impaired by increased calcium concentrations [63]. A well-documented biological effect of ANG II signaling via AGTR2 is vasodilation, which is highlighted by our transcriptome data [13,46].

Transcriptome analysis revealed 94 losartan-induced genes associated with diverse biological processes and pathways such as vascular smooth muscle contraction, the oxytocin signaling pathway, renin secretion, and ECM-receptor interaction. Co-stimulation with losartan and ANG II induced exclusive expression of 106 genes associated with extracellular matrix organization, metal ion transport, DNA methylation or demethylation, cell differentiation, the developmental process, response to muscle stretch, calcium ion transmembrane transport, and the regulation of cytokine production involved in the inflammatory response.

These findings highlight the usefulness of human urine-derived podocytes in studying nephrogenesis, kidney-associated diseases such as hypertension, the RAAS pathway, nephrotoxicity, and drug screening. Figure 5 presents a graphical summary of this study.

## 4. Materials and Methods

### 4.1. Cell Culture Conditions

Urine-derived renal progenitor cells (UdRPCs) of a 51-year-old male (UM51) were isolated as described in Rahman et al. [24]. Further, the UM51 cell line was immortalized with an hTERT-expressing plasmid (UM51 hTERT) [28]. The cell lines were cultured on 0.2% type 1 collagen (Thermo Fisher Scientific, Waltham, MA, USA) coated 6- or 12-well plates at 37 °C under hypoxic conditions. The cells were cultured in Proliferation Medium (PM) composed of 50% DMEM high-glucose (Gibco) and 50% keratinocyte growth basal medium (Lonza, Basel, Switzeland) supplemented with 5% fetal bovine serum (Gibco), 0.5% Non-Essential Amino Acid (Gibco), 0.25% Glutamax (Gibco), and 0.5% penicillin and streptomycin (Gibco). For further differentiation into podocytes, the cells were seeded at a low density (50,000 cells per 6-well) and cultured for 24 h in PM. On the next day, the medium was exchanged to Advanced RPMI 1640 (Gibco) supplemented with 0.5% fetal bovine serum, 1% penicillin and streptomycin, and 30 μM retinoic acid (Sigma-Aldrich Chemistry, Steinheim, Germany). After 7 days, the typical podocyte morphology was observed. Losartan (Sigma-Aldrich Chemistry) and ANG II (Sigma-Aldrich Chemistry) were diluted using Advanced RPMI 1640 to a final concentration of 100 μM ANG II or 1 μM losartan. First, the cells were incubated for 24 h with 1 μM losartan and then for 24 h with ANG II.

### 4.2. Immunofluorescence Staining

For the immunocytochemistry, the podocytes were fixed with 4% paraformaldehyde (PFA) (Polysciences, Warrington, FL, USA) followed by three wash steps with PBS for 5 min each. To block unspecific binding sites, the fixed cells were incubated with blocking buffer containing 10% normal goat or donkey serum, 1% BSA, 0.5% Triton X-100 (Sigma-Aldrich Chemistry) and 0.05% Tween-20 (Sigma-Aldrich Chemistry) for 2 h at room temperature. The primary antibodies were diluted in blocking buffer according to Appendix A and incubated at 4 °C overnight. On the following day, the cells were washed once with 0.05% Tween-20 diluted in PBS and two times with PBS only. After washing them, a fluorochrome conjugated secondary antibody together with HOECHST (1:5000) (Thermo Fisher Scientific, Waltham, MA, USA) were added and incubated for 1 h at room temperature protected from light. The secondary antibodies were diluted in blocking buffer according to Appendix A. After three consecutive washing steps, the visualization was performed by using a fluorescence microscope (LSM700).

### 4.3. Relative Quantification of Podocyte-Associated Gene Expression via Real-Time PCR

Quantitative PCR was performed using podocyte-specific genes and angiotensin II receptors of type I and II. The samples were run in triplicate on a 384-well reaction plate and Power Sybr Green PCR Master Mix (Applied Biosystems, Foster City, CA, USA) using Step One Plus Real-Time PCR systems. The amplification conditions were denaturation at 95 °C for 13 min followed by 37 cycles of 95 °C for 50 s, 60 °C for 45 s, and 72 °C for 30 s. To normalize the quantitative real-time PCR, the ribosomal encoding gene-RPL0 was used. The primer sequences are listed in Appendix A. The results were analyzed using the 2^−∆∆CT^ method and specified by fold change expression. The samples were run in independent experiments three times. The mean value of these experiments was used for the calculations. The statistical significance was calculated using the Two-Sample test assuming unequal variances [65] with a significance threshold *p* = 0.05.

### 4.4. Western Blot Analysis

The podocytes were lysed in RIPA buffer (Sigma-Aldrich Chemistry) containing 5 M NaCl, 1% NP-40, 0.5% DOC, 0.1% SDS, 1 mM EDTA, 50 mM Tris, pH 8.0, and freshly added 10 μL/mL protease and phosphatase inhibitor (Sigma-Aldrich). We dissolved 20 μg of each obtained protein lysate in a 10% sodium dodecyl sulfate PAGE gel and then transferred it to an Immobilon-P membrane (Merck Millipore, Burlington, VT, USA). The membranes were incubated with the primary antibody overnight at 4 °C, then washed three times with 0.1% Tween-20 in Tris-buffered saline followed by 1 h of incubation of the secondary antibodies at room temperature. For visualization of the blotted proteins, Pierce^TM^ ECL Western Blotting Substrate solutions from Thermofisher (Thermo Fisher Scientific, Waltham, MA, USA) were used. The quantification was carried out with ImageJ. Detailed information on the antibodies used can be found in Appendix A.

### 4.5. Microarray Data Analyses

For the microarray experiments, 1 μg RNA preparations were hybridized on the Human Clariom S Gene Expression Array (Affymetrix, Thermo Fisher Scientific, Waltham, MA, USA) at the core facility Biomedizinisches Forschungszentrum (BMFZ) of the Heinrich Heine University Düsseldorf. The raw data were imported into the R/Bioconductor environment [66] and further processed with the package oligo [67] using background correction, logarithmic (base 2) transformation, and normalization with the robust multi-array average (RMA) method. Venn diagrams were generated with the Venn diagram package. Subsets from the Venn diagrams were used for follow-up GO and pathway analyses [68]. Heatmaps were drawn with the R package gplots [69] using Pearson correlation as similarity measure and color scaling per gene.

Gene expression data will be available online in the National Centre of Biotechnology Information (NCBI) Gene Expression Omnibus. 

### 4.6. Ethics Statement

In this study, urine samples were collected with the informed consent of the donors and the written approval (Ethical approval Number: 5704) of the ethical review board of the medical faculty of Heinrich Heine University, DuȠsseldorf, Germany. All methods were carried out in accordance with the approved guidelines. The medical faculty of Heinrich Heine University approved all experimental protocols. 

### 4.7. Statistics

Data are presented as arithmetic means + standard error. In total, three independent experiments were performed and used for the calculation of mean values. The statistical significance was calculated using the two-sample Student’s *t*-test with a significance threshold *p* = 0.05.

## Figures and Tables

**Figure 1 ijms-24-10551-f001:**
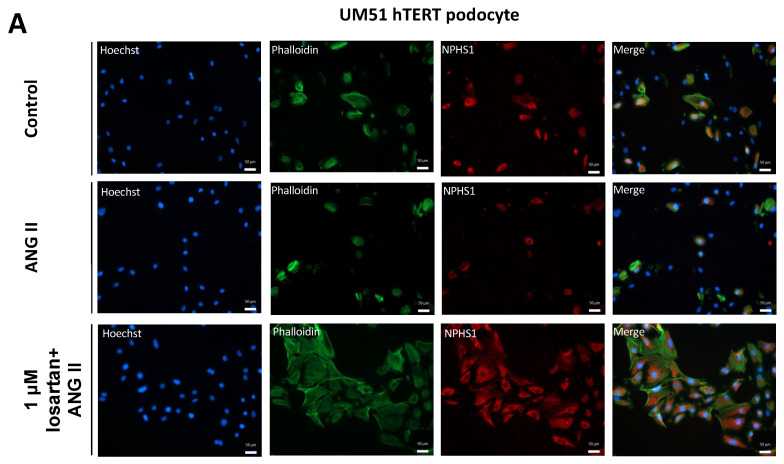
Losartan enhances NPHS1 expression in UdRPC-derived podocytes. UM51 and immortalized UM51-hTERT were differentiated into podocytes by culturing at 60–70% density in advanced RPMI medium supplemented with 30 μM RA. Podocytes were pre-treated with 1 μM losartan for 24 h and subsequently treated with 100 μM ANG II for a further 24 h. NPHS1 was expressed at both losartan concentrations and some cells adopted a rounded morphology (*n* = 3). (**A**) corresponds to the UM51 hTERT cell line and (**B**) the primary UM51 podocytes.

**Figure 2 ijms-24-10551-f002:**
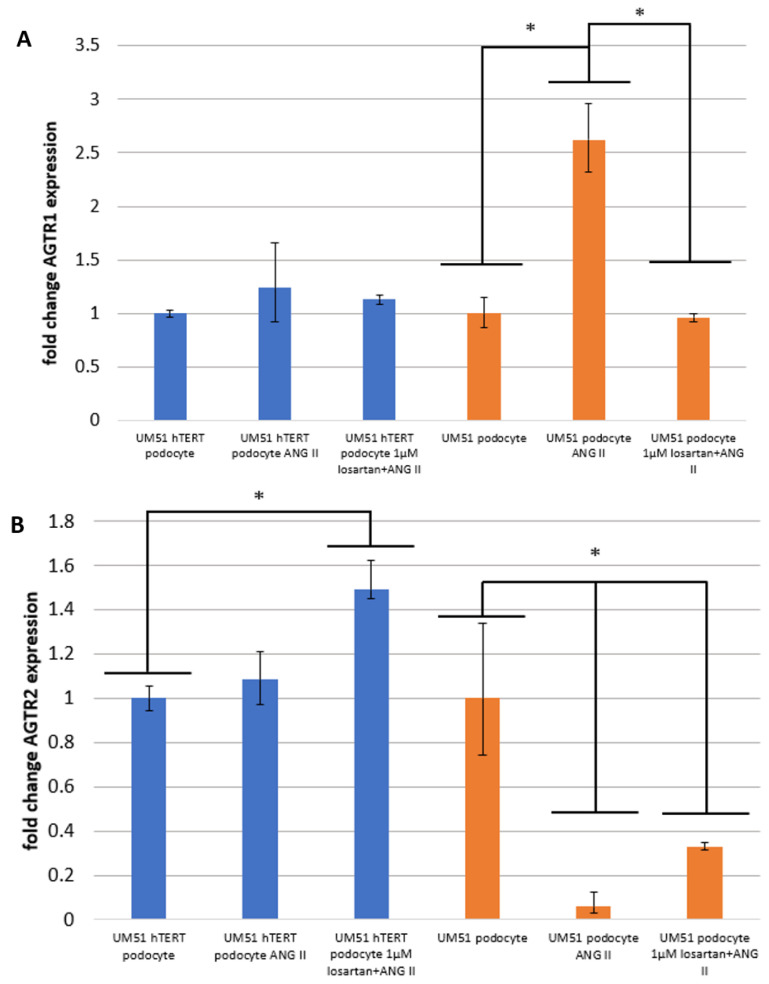
Stimulation of podocytes with losartan and Ang II induced changes in the expression of angiotensin receptors I and II. Urine-derived renal progenitor cells UM51 and immortalized UM51-hTERT were differentiated into podocytes. The RAAS-associated receptors AGTR1 (**A**,**D**), AGTR2 (**B**,**D**), and NPHS1(**C**,**D**) were analyzed via qPCR (**A**–**C**) and Western blotting (**D**). Expression of Glyceraldehyde-3-phosphate dehydrogenase (GAPDH) was used for normalization. Full uncropped Western blot images are presented in Appendix A. To normalize the quantitative real-time PCR, the ribosomal encoding gene RPL0 was used. Primary UM51 podocytes are depicted in orange and the immortalized UM51-hTERT podocytes are represented in blue (*n* = 3). The statistical significance was calculated using the two-sample Student’s *t*-test with a significance threshold *p* = 0.05 is marked by an asterisk (*).

**Figure 3 ijms-24-10551-f003:**
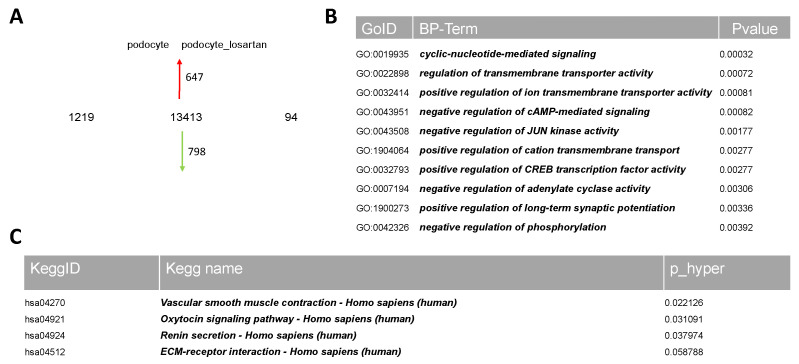
Transcriptome analysis of urine-derived podocytes treated with and without losartan. Comparative transcriptome and Gene Ontology analysis of urine-derived UM51 podocytes treated with and without 1 μM losartan. Expressed genes (det- *p* < 0.05) in UM51 derived podocytes treated with and without 1 μM losartan compared in the Venn diagram (**A**) show distinct (1219 in untreated podocytes and 94 in podocytes treated with 1 μM losartan) gene expression patterns. In total, 13,413 genes were found to be expressed in common between untreated podocytes and podocytes treated with 1 μM losartan. Podocytes treated with 1 μM losartan showed the most over-represented GO BP-terms associated with regulation of monoatomic ion transmembrane transporter activity, cell morphogenesis involved in neuron differentiation, and cell–cell communication (**B**,**D**). The 94 significant genes associated with KEGG pathways revealed vascular smooth muscle contraction, oxytocin signaling pathway, renin secretion, and ECM-receptor interaction (**C**).

**Figure 4 ijms-24-10551-f004:**
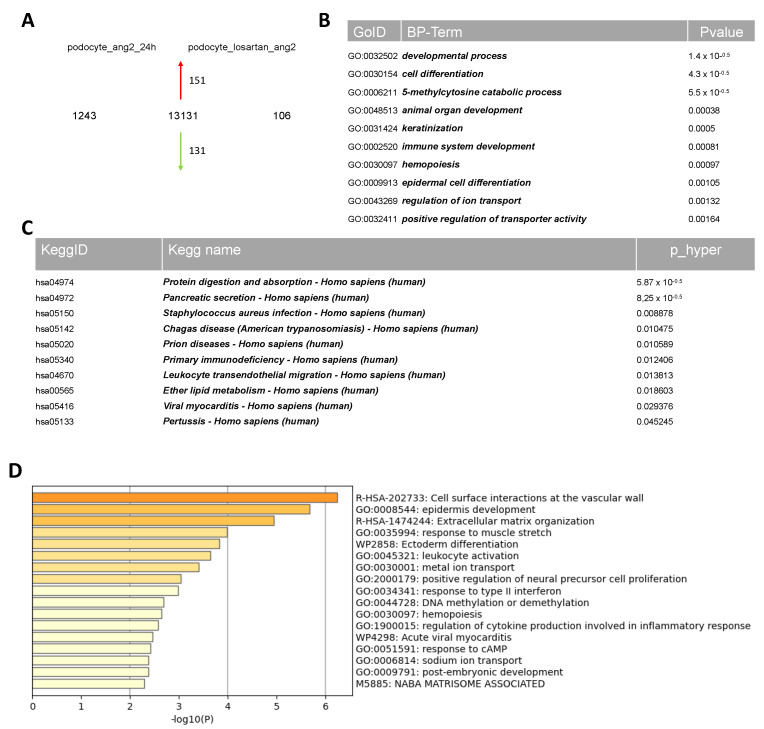
Transcriptome analysis of urine-derived podocytes stimulated with ANG II and losartan. Comparative transcriptome and Gene Ontology analysis of urine-derived UM51 podocytes stimulated with 100 μM ANG II with and without 1 μM losartan. Expressed genes (det-*p* < 0.05) in UM51 derived podocytes treated with 100 μM ANG II and with and without 1 μM losartan compared in the Venn diagram (**A**) show distinct (1243 in podocytes only treated with 100 μM ANG II; 106 in podocytes treated with the combination of 100 μM ANG II and 1μM losartan) gene expression patterns. We found 13,131 genes to be expressed in common between podocytes treated with 100 μM ANG II and podocytes treated with the combination of 100 μM ANG II and 1 μM losartan. Podocytes treated with the combination of 100 μM ANG II and 1 μM losartan showed the most over-represented GO BP-terms associated with extracellular matrix organization, metal ion transport, DNA methylation or demethylation, and regulation of cytokine production involved in inflammatory response and response to cAMP (**B**,**D**). KEGG pathway analyses revealed, for example, protein digestion and absorption, pancreatic secretion, and primary immunodeficiency (**C**).

**Figure 5 ijms-24-10551-f005:**
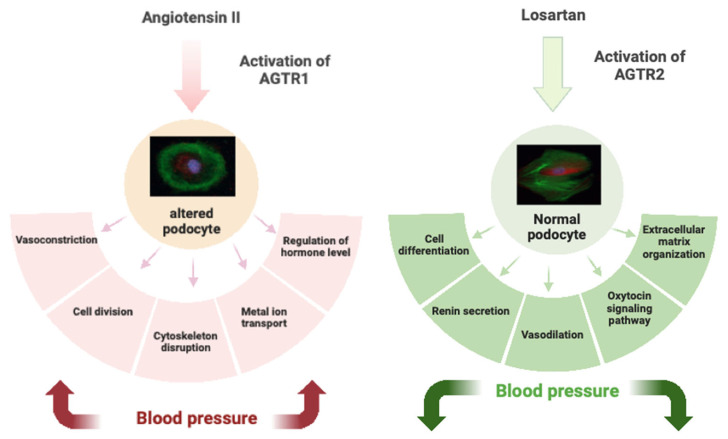
Biological consequences of ANG II binding on angiotensin II type I and II receptors. The renin–angiotensin–aldosterone system (RAAS) is a bio-enzymatic cascade that contributes to various systematic and localized processes. The octapeptide angiotensin II (ANG II) is the biological effect of the RAAS and regulates blood pressure and extracellular volume. ANG II can bind two types of receptors that are involved in the regulation of renal hemodynamics and tubular function. These effects are dependent on binding to the two subtypes of receptors angiotensin receptors type I (AGTR1) and angiotensin receptors type II (AGTR2). ANGII binding and signaling through AGTR1 is associated with cell division, cytoskeleton disruption, vasoconstriction, metal ion transport, and regulation of hormone levels, e.g., estrogen or cholecystokinin. The development of orally active, non-peptide, selective AGTR1 blockers began with the synthesis of losartan. Preventing the binding of ANGII to AGTR1 leads to binding and signaling through AGTR2, which is associated with vasodilation, renin secretion, cell differentiation, extracellular matrix organization, and activation of the oxytocin signaling pathway [64].

## Data Availability

Not applicable.

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
