# Peer review of "Unveiling Angiotensin II and Losartan-Induced Gene Regulatory Networks Using Human Urine-Derived Podocytes"

_ijms, 2023, doi:10.3390/ijms241310551_

Round 1

Reviewer 1 Report

The authors described the effects of losartan treatment on the molecular changes induced by ANG II on urine-renal progenitors using Western blot, RT-PCR, immunofluorescence and transcriptome analysis.

The main point that needs to be revised concers the number of experiments performed by the researchers. This data is not present neither in the figure legends or in the materials and methods sections. From the figures it seeems that only one experiment has been performed and showed. The authors must report the number of the experiments considering that n=1 is not statically relevant. Furthermore, the Western plot in  figure 2 D on hTERT podocytes shows some critical points, such as absence of bands  or bands of different molecolar weights (NPHS1 in lanes 1,  2 and 3???; AGTR1 in lane 3???; GAPDH in lane 3???). Similarly, in the western plots showed in supplementary figure 1, it is hard to understand which lane of which plot corresponds to the data presented in main figure of the manuscript. These are important points that needs revision. 

Reviewer 2 Report

The manuscript “Unveiling angiotensin II and losartan- induced gene regulatory 2 networks using human urine-derived podocytes” by Thimm et al. is an interesting work investigating the effects of Angiontensin II (AII) with or without losartan on human podocytes derived from SIX2-positive urine renal progenitor cells (UdRPCs), which were immortalized or not. The authors analysed in these cells the expression of AII receptors (AGTR1 and 2) and performed also a transcriptome analysis of the human podocytes under the different incubation conditions.

The work is of interest; however, I have some question about it.

General comments

1) The authors described the presence of AII receptors in different segments of the nephron. However, I have found no information about their expression in podocytes. Can the authors comment on this? Which AII receptors are expressed in human podocytes?

2) The authors performed their experiments using a AII concentration of 100 µM and a losartan concentration of 1 µM. Such concentrations seem to me to be very high, considering that the circulating levels of Angiotensin II are in the pM-nM range (Clin Proteom 11, 37 (2014), https://doi.org/10.1186/1559-0275-11-37 and JRAAS 2001;2 (suppl 1):S176-S184) and interaction of losartan with ATR1 is in the nM range (The Lancet, Volume 355, Issue 9204, 2000, Pages 637-645, https://doi.org/10.1016/S0140-6736(99)10365-9). Perhaps the authors can better explain why they used these concentrations.

Specific comments

3) Line 103: The authors stated that: “To date transcriptome data derived from human podocytes treated with losartan are restricted to rodents”. How can data from human podocytes be restricted to rodents? Please correct this sentence.

4) Lines 148 and 156: I think the authors mean p < 0.05 and not p> 0.05, since they are discussing statistically significant effects.

5) Figure 1. The authors state that the cells were differentiated to podocytes. What are the parameters that are leading to this statement? Which podocytes-typical characteristics display these cells? Do the cells form foot processes when cultivated? The labelling for nephrin seems to me to be a little bit strange, since it shows nephrin to be localized in intracellular compartments and not in the plasma membrane, as one would expect in podocytes. Are the authors sure that the antibody used in the immunofluorescence analysis is working properly?

6) Figure 2 A-C. In these panels I suggest indicating (for example using an asterisk) which columns are statistically significant different. What type of error is indicated in these panels? SD? SEM? Please specify.

7) Regarding the statistical analysis of the results, I suggest adding a paragraph under Material and Methods, where the statistical methods used in this work are explained (t-test, Anova, etc.).

8) Figure 2D: why the nephrin band in the lane “UM51hTERT podocyte 1 μM losartan+ANG II“ has a different molecular weight than the other two presented? Also a part of the nephrin band in the lane “UM51hTERT podocyte+ANG II“ has been cut away. Why there are no bands for AGTR1 and for GAPDH in the lane “UM51hTERT podocyte 1 μM losartan+ANG II“?  Also it is very difficult to follow the labelling of the different lanes looking at the images of uncropped Western blot analysis presented in the Supplementary Material. Here a labelling of the different lanes would help to understand what the authors are showing. Since multiple bands for the protein under investigation are visible, I wonder which one is specific and why.

9) Figure 2D: why there is no error (SD or SEM) in the Western blot quantification? Did the authors performed the Western blot analysis only once?

10) line 215: I think that “Of these” is not appropriate, since they are not co-expressed.

11) I miss the citation and discussion of the publication “Angiotensin II regulates phosphorylation of actin-associated proteins in human podocytes” FASEB J. 2017 Nov;31(11):5019-5035. doi: 10.1096/fj.201700142R, which addressed a very similar question in human podocytes.

Round 2

Reviewer 1 Report

Following the revision the manuscript is acceptable for publication.

Reviewer 2 Report

I thank the authors for the satifactory answers to my questions.